# Convalescent Plasma Treatment of Patients Previously Treated with B-Cell-Depleting Monoclonal Antibodies Suffering COVID-19 Is Associated with Reduced Re-Admission Rates

**DOI:** 10.3390/v15030756

**Published:** 2023-03-15

**Authors:** Petros Ioannou, Athanasios Katsigiannis, Ioanna Papakitsou, Ioannis Kopidakis, Eirini Makraki, Dimitris Milonas, Theodosios D. Filippatos, George Sourvinos, Marina Papadogiannaki, Evaggelia Lydaki, Georgios Chamilos, Diamantis P. Kofteridis

**Affiliations:** 1School of Medicine, University of Crete, 71003 Heraklion, Greecehamilos@uoc.gr (G.C.); 2COVID-19 Department, University Hospital of Heraklion, 71110 Heraklion, Greece; 3Laboratory of Clinical Virology, School of Medicine, University of Crete, 71003 Heraklion, Greece; sourvino@med.uoc.gr; 4Department of Blood Transfusion, University Hospital of Heraklion, 71110 Heraklion, Greece; mpapadog2004@yahoo.gr (M.P.);; 5Microbiology Department, University Hospital of Heraklion, 71110 Heraklion, Greece

**Keywords:** anti-CD20, rituximab, Obinutuzumab, COVID-19, SARS-CoV-2, lymphoma, convalescent plasma

## Abstract

Patients receiving treatment with B-cell-depleting monoclonal antibodies, such as anti-CD20 monoclonal antibodies, such as rituximab and obinutuzumab, either for hematological disease or another diagnosis, such as a rheumatological disease, are at an increased risk for medical complications and mortality from COVID-19. Since inconsistencies persist regarding the use of convalescent plasma (CP), especially in the vulnerable patient population that has received previous treatment with B-cell-depleting monoclonal antibodies, further studies should be performed in thisdirection. The aim of the present study was to describe the characteristics of patients with previous use of B-cell-depleting monoclonal antibodies and describe the potential beneficial effects of CP use in terms of mortality, ICU admission and disease relapse. In this retrospective cohort study, 39 patients with previous use of B-cell-depleting monoclonal antibodies hospitalized in the COVID-19 department of a tertiary hospital in Greece were recorded and evaluated. The mean age was 66.3 years and 51.3% were male. Regarding treatment for COVID-19, remdesivir was used in 89.7%, corticosteroids in 94.9% and CP in 53.8%. In-hospital mortality was 15.4%. Patients who died were more likely to need ICU admission and also had a trend towards a longer hospital stay, even though the last did not reach statistical significance. Patients treated with CP had a lower re-admission rate for COVID-19 after discharge. Further studies should be performed to identify the role of CP in patients with treatment with B-cell-depleting monoclonal antibodies suffering from COVID-19.

## 1. Introduction

Coronavirus disease-19 (COVID-19) has affected more than 676,000,000 people, most commonly causing asymptomatic disease, but has caused more than 6,770,000 deaths, affecting more commonly and more severely patients with medical co-morbidities, such as those with hematological malignancies [1,2,3]. More specifically, despite available treatment options, patients with hematological diseases are at an increased risk of complications and death, especially if the underlying disease was leukemia as shown in the EPICOVIDEHA study [3]. Despite the reducing mortality rates with the next COVID-19 waves after the initial variant and the availability of the vaccines, COVID-19 poses a significant threat to this patient population [4]. More specifically, patients receiving treatment with B-cell-depleting monoclonal antibodies, such as anti-CD20 monoclonal antibodies, such as rituximab and obinutuzumab, either for hematological disease or another diagnosis, such as a rheumatological disease, are at an increased risk for medical complications and mortality from COVID-19 [5]. Furthermore, depletion of B-cells by using monoclonal antibodies is associated with hypogammaglobulinemia and reduced antibody response to COVID-19, and could be associated with relapsing COVID-19 [6,7].

Due to the limited therapeutic approaches early during the pandemic and the difficulties of medical management of patients with COVID-19 that were receiving treatment with B-cell-depleting monoclonal antibodies, several attempts were made to supplement standard medical treatment with therapies that could partially restore the particular immunodeficiency noted in this patient population. For example, in a relatively recent study in 180 B-cell-depleted patients with COVID-19, rates of progression to severe disease, hospitalization, admission to the intensive care unit (ICU), mortality as well as persistent COVID-19 were lower in patients treated with anti-spike monoclonal antibodies [8]. On the other hand, convalescent plasma (CP) therapy has been used in the past with partial success in specific viral outbreaks, as in the case of the Spanish flu in 1918, and more recently in the SARS pandemic caused by SARS-CoV-1 in 2003 [9,10]. Administering blood plasma collected from patients that have recovered from COVID-19 to newly symptomatic patients has been shown to be safe and may reduce viral load and lead to clinical improvement [10,11,12,13]. However, a more recent study did not find any difference in the clinical status or overall mortality among patients treated with CP and those that received placebo [14]. A more recent study regarding early outpatient treatment for COVID-19 with CP showed that early administration, within nine days after the onset of symptoms, may lead to reduced rates of disease progression in adult patients regardless of risk factors for disease progression or vaccination status [15]. Thus, since inconsistencies persist regarding the use of CP, especially in the vulnerable patient population that has received previous treatment with B-cell-depleting monoclonal antibodies, further studies should be performed to that direction.

The aim of the present study was to describe the characteristics of patients with previous use of B-cell-depleting monoclonal antibodies and describe the potential beneficial effects of CP use in terms of mortality, ICU admission and disease relapse.

## 2. Materials and Methods

### 2.1. Study Population

This is a retrospective cohort study. Study participants were patients admitted to the COVID-19 Department of the University Hospital of Heraklion, Greece, until December 2022. Patients were included if they were adults, were diagnosed with COVID-19 with a positive Reverse Transcription-Polymerase Chain Reaction (RT-PCR) test for Severe Acute Respiratory Syndrome coronavirus 2 (SARS-CoV-2) and were admitted to the COVID-19 Department. Exclusion criteria were the absence of a positive RT-PCR test for SARS-CoV-2 and the transfer of patients to another hospital that did not allow the completion of data collection. Data recorded and evaluated included age, gender, clinical and radiological as well as laboratory characteristics, treatment administered and outcomes.

As COVID-19 CP, we defined plasma collected from patients that have recovered from COVID-19 and contained antibodies against SARS-CoV-2. The concentration of anti-SARS-CoV-2 IgG against receptor binding domain (RBD) of spike protein S1 was measured by enzyme-linked immunosorbent assay (ELISA) (EI-2606-9601 G ELISA kit, EUROIMMUN, Lübeck, Germany). The strong positive relationship between anti-S1 IgG levels and neutralizing antibody titers has been previously demonstrated [16], substantiating the implementation of the QuantiVac ELISA of Euroimmun to assess protective immunity following infection. More specifically, the Department of Blood Transfusion of the University Hospital of Heraklion was responsible for the CP donors’ information and CP collection, processing and storage. Blood donors were categorized as having high or low titers, with a cutoff of >3 (with SARS-CoV-2 antibody detection in the blood of CP donors by ELISA characterized as negative for an IgG ratio of <0.8, borderline for a ratio of 0.8–1.1, and positive for a ratio of >1.1). Blood was collected from donors during all periods of the pandemic to allow collection of CP with activity against the variants of concern that were prevalent at each time. Technically, the plasma was collected by plasmapheresis on Trima Accel^®^ Automated Blood Collection System V 6.0 (Caridian BCT, Lakewood, CO, USA). The volume of each CP collection was 600 or 800 mL, according to the total blood volume of the donor. Plasma obtained by plasmapheresis was split before freezing into three to four separate units so that each unit for transfusion was 200 mL. Pathogen inactivation was performed immediately after CP collection using the INTERCEPT Blood System (Cerus Corporation, Concord, CA, USA). The CPs were stored in a freezer at −80 °C until use. The inclusion criteria for donation by the donors were the following:Patients (symptomatic or asymptomatic) diagnosed with COVID-19, confirmed by molecular technique (RT-PCR).The CP apheresis started at least 14 days after complete recovery or 14 days after a negative nasopharyngeal swab testing for the presence of SARS-CoV-2 RNA and with the donor having two negative tests within 24 h.The donors fulfilled the general criteria for whole blood donation or apheresis (age between 18 and 60 years, hemoglobin ≥12.5 g/dL for women and ≥13.5 g/dL for men, heart rate between 50 and 100/min with no irregularities, blood pressure—systolic 100–180 mm Hg and diastolic 50–100 mm Hg, temperature < 37.5 °C, body weight ≥ 50 kg and donor being in a healthy general condition both physically and mentally.In addition, the CP donors were required not to have a history of blood transfusion and women CP donors were required not to have a history of pregnancy. In other cases, tests for the presence of anti-HLA antibodies (anti-HLA), anti-human neutrophil antigens antibodies (anti-HNA) and anti-human platelet antigens antibodies (anti-HPA) were performed and were required to be negative.Donors were tested with ELISA for the presence of antibodies against SARS-CoV-2 and were also tested with serology and molecular testing for blood-borne pathogens (HIV, HBV, HCV, HTLV, syphilis) which was required to be negative.

The study was approved by the Ethics Committee of the University Hospital of Heraklion.

### 2.2. Statistics

Categorical data were analyzed with Fisher’s exact test. Continuous variables were compared using Student’s t-test for normally distributed variables and the Mann–Whitney U-test for non-normally distributed variables. All tests were two-tailed and *p* ≤ 0.05 was considered to be significant. Data are presented as numbers (%) for categorical variables and medians (interquartile range (IQR)) or means (±standard deviation (SD)) for continuous variables. A linear regression analysis model was developed to evaluate the effect of several parameters (age, gender, clinical, radiological and laboratory characteristics as well as treatment administered) with mortality. All the parameters mentioned above were calculated with GraphPad Prism 6.0 (GraphPad Software, Inc., San Diego, CA, USA). A multivariate logistic regression analysis model was developed to evaluate the association of factors identified in the univariate analysis with a *p* ≤ 0.05 with mortality. Multivariate analysis was performed using the SPSS version 23.0 (IBM Corp., Armonk, NY, USA).

## 3. Results

During the study period, 39 patients with previous use of B-cell-depleting monoclonal antibodies were hospitalized in the COVID-19 department. The most common monoclonal antibody used was rituximab in 94.9% (37 patients) and obinutuzumab in 5.1% (2 patients). The underlying diseases necessitating B-cell-depleting monoclonal antibody use was hematological disease in 56.4% (22 patients) and rheumatological disease in 38.5% (15 patients). The mean age was 66.3 years and 51.3% (20 patients) were male. Fever was present in 66.7% (26 patients) and 27% (10 patients) presented with respiratory failure. Infiltrates were present in the chest X-rays in 61.5% (24 patients). Regarding treatment for COVID-19, remdesivir was used in 89.7% (35 patients), corticosteroids in 94.9% (37 patients) while antibacterials were used in 89.7% (35 patients), even though a co-infection was demonstrated in 10.3% (4 patients). CP was administered in 53.8% (21 patients). The mean time between plasma collection and plasma use was 38 days. All the CPs that were used in the present study had an ELISA value of 3 or higher. The maximum ELISA value of the donors was 10.57 and the median ELISA value for the donors was 6.65. ICU was needed for 15.4% (6 patients), while in-hospital mortality was 15.4% (6 patients) in total. No patient received tociluzumab or baricitinib. Five patients among the survivors had a persistently positive PCR for SARS-CoV-2 for up to at least three months after the initial diagnosis and were considered to have a chronic infection. Table 1 shows the characteristics of all patients and a comparison of the patients’ characteristics among those who were treated with CP and those who were not. Patients who received CP had a lower likelihood of re-admission for COVID-19 and a trend towards less need for ICU admission, even though the last did not reach statistical significance.

Patients who died had a higher age; however, this did not reach statistical significance. Furthermore, patients who died were more likely to have hematological disease as a cause of treatment with B-cell-depleting monoclonal antibodies, were more likely to need ICU admission and also had a trend towards a longer hospital stay, even though the last did not reach statistical significance. Table 2 shows a comparison of the characteristics among patients who survived with those who died during the hospitalization. Cause of death was respiratory failure in three patients (in two of them, respiratory failure developed later in the course of hospitalization) and septic shock in the remaining three patients. In patients with septic shock, cultures were negative in two patients, but in one patient, *Enteroccoccus faecalis* and *Pseudomonas aeruginosa* were isolated in a culture from pus, and were considered the causative pathogens. All patients who died, died during their first admission for COVID-19.

To identify factors associated with in-hospital mortality among patients with COVID-19 and previous use of B-cell-depleting monoclonal antibody use, we performed regression analysis. First of all, a linear regression analysis was performed and identified higher lymphocyte number (*p* = 0.0387), longer duration of stay (*p* = 0.0021) and need for ICU admission (*p* = 0.0097) to be independently associated with mortality. However, a multivariate logistic regression analysis identified only a longer duration of stay to be independently associated with increased likelihood of in-hospital mortality (*p* = 0.042) with an OR (per day) of 1.08 (95% CI 1.003–1.163). Table 3 shows the results of the regression analysis of in-hospital mortality in patients with COVID-19 and previous B-cell-depleting monoclonal antibody use.

To identify factors associated with readmission among patients with COVID-19 and previous use of B-cell-depleting monoclonal antibody use who survived the hospitalization, we performed another regression analysis. First of all, a linear regression analysis was performed and identified higher lymphocyte number (*p* = 0.0012) and CP use (*p* = 0.022) to be independently associated with reduced re-admission rates for COVID-19. However, a multivariate logistic regression analysis failed to identify factors independently associated with reduced readmission for COVID-19 among patients who survived the index hospitalization for COVID-19. Table 4 shows the results of the regression analysis of re-admission rates in patients with COVID-19 and previous use of B-cell-depleting monoclonal antibody use.

## 4. Discussions

Herein, we present data from a retrospective cohort of patients with previous use of B-cell-depleting monoclonal antibodies that were admitted with COVID-19 in a tertiary hospital. Mortality was 15.4%, while patients who died were more likely to need ICU admission and also had a trend towards a longer hospital stay, even though the last did not reach statistical significance. Patients treated with CP had a lower re-admission rate for COVID-19 after discharge.

COVID-19 has led to hundreds of millions of infections and has caused millions of deaths, putting an enormous pressure on healthcare and society [17,18,19]. Furthermore, the post-COVID-19 has come to add to the already increased burden of COVID-19, as long-term consequences of SARS-CoV-2 infection are increasingly recognized in patients with COVID-19 involving respiratory, cardiovascular and neuropsychiatric symptoms as well as fatigue, thus affecting patients’ lives [20,21]. However, not all patients with COVID-19 are equally susceptible to acquiring disease, as they may carry different risks for mortality and development of COVID-19-related complications after the disease [22,23,24,25]. For example, patients with underlying immunosuppression, such as patients with cancer or hematological disease, as well as other patients with drug-induced immunosuppression, are more likely to suffer with severe COVID-19, necessitate hospitalization and may also have higher mortality [26,27,28,29,30].

Therapy with B-cell-depleting monoclonal antibodies has revolutionized the treatment of several diseases, such as hematological diseases associated with antibody production or B-cell malignancies, rheumatological diseases and other autoimmune diseases, such as multiple sclerosis [31,32,33,34,35,36,37,38]. However, therapy with these monoclonal antibodies, such as rituximab or obinutuzumab, has been associated with hypogammaglobulinemia and impaired reconstitution of B-cells as well as a higher rate of infections [39]. Immune response against viral pathogens is mainly mediated by T-cell lymphocytes and natural killer (NK) cells. B-cells constitute a critical component of the humoral immunity, by producing antigen-specific neutralizing monoclonal antibodies, while they also serve as professional antigen-presenting cells, thus inducing antigen-specific T-cell activation and differentiation [40]. Thus, use of B-cell-depleting monoclonal antibodies is anticipated to reduce the host’s ability to clear viral antigens due to reduced antibody production and reduced B-cell-dependent T-cell activation. Indeed, there are several reports of patients previously treated with B-cell-depleting monoclonal antibodies who presented with severe and protracted COVID-19 [41,42].

Herein, we retrospectively studied a cohort of patients admitted to the COVID-19 department of the University Hospital of Heraklion who had been previously treated with a B-cell-depleting monoclonal antibody and identified a mortality of about 15.4% which is a rate similar to that for the general population hospitalized for COVID-19 [43,44,45,46]. Notably, the mortality noted in patients with previous use of B-cell-depleting monoclonal antibodies was also similar to the mortality in the general population noted in another recent study involving patients hospitalized in the same department as in the present study [47].

Mortality in the present cohort was higher among patients with hematological disease as the cause of previous treatment with a B-cell-depleting monoclonal antibody. This is reasonable, since these patients usually receive these treatments due to B-cell malignancies and may also receive other immunocompromising treatments along with the B-cell-depleting monoclonal antibody, leading to an increased overall risk for mortality [48].

Moreover, the duration of hospitalization was also higher among patients who died in the present study. Furthermore, duration of stay in the hospital was the only factor that was identified to be independently associated with in-hospital mortality. This is a relatively known concept, as increased duration of hospitalization has been linked with increased mortality in other studies and medical conditions as well [49].

The concept of passive immunotherapy was introduced during the 19th century as a therapy against viral infections. In the current era, it poses a notable treatment option against COVID-19. Administration of CP has best proven efficacy in patients older than 65 years and within 3 days of symptom onset, where a reduction of progression to severe respiratory disease was found to be as high as 48% [50]. Therapy with plasma has several benefits, such as antibody-mediated viral suppression, viral clearance and increased infected-cell clearance via activation of the complement, phagocytosis and antibody-dependent cellular cytotoxicity [51]. These imply that CP use in patients with previous use of B-cell-depleting monoclonal antibodies may serve as a valuable therapeutic modality.

The use of CP in patients with previous use B-cell-depleting monoclonal antibodies presenting with severe COVID-19 may be associated with improved clinical outcomes and reduced mortality, as also suggested by the literature [9,52,53,54]. Furthermore, early CP administration could also be considered in severely immunocompromised patients with previous B-cell-depleting monoclonal antibody use that present early with mild COVID-19 in order to prevent progression to severe COVID-19. There are studies suggesting that early CP use in mildly ill infected individuals could be associated with a reduced risk of progression to severe COVID-19, as has been shown in older individuals [55]. Even though there are reports suggesting that this practice may not be beneficial, as in the study published by Korley et al. [56], there are some specific studies in severely immunocompromised individuals that may suggest benefit with CP use [57,58,59,60]. Indeed, in the study be Korley et al., immunocompromised patients were less than 13% of the patients that were treated with CP, and thus, the results of this study may not be applicable to the population described in the current study [56]. However, a recent living systematic review evaluating the question of whether treatment with CP is beneficial in terms of mortality in patients with COVID-19 concludes that the effect in severe disease is probably minimal, but it is unclear whether it could be beneficial in mild COVID-19 and suggests that further studies on the topic may be needed [61]. In the present study, there was no difference in mortality between patients who were treated with CP and those that were not. However, there was a statistically significant difference among survivors in regard to future hospitalization for the same reason, with patients who were treated with CP having a lower rate of hospitalization for COVID-19. There are studies suggesting that patients with previous treatment with B-cell-depleting monoclonal antibodies may have a protracted course of symptoms [41]. In these patients, treatment with CP could allow for faster elimination of the virus, allowing for a faster recovery [53,62]. To that end, even though treatment with CP may not affect mortality in these patients, the possibility for faster elimination of the virus, reduction in the possibility of future admissions for COVID-19 and the possibility for reduction in the severity of remaining symptoms, even though this was not directly assessed in the present study, could justify the use of CP in this patient population. Thus, we feel that further studies should be performed to investigate the role of CP in patients with previous treatment with B-cell-depleting monoclonal antibodies and COVID-19, with a focus in outcomes other than mortality, such as future readmissions for COVID-19, long-COVID-19, post-COVID-19 and quality of life.

Even though treatment with CP is considered safe, and no adverse effects were noted in the small population of this study, treatment with plasma may cause some notable adverse reactions, such as allergic and anaphylactic reactions, hemolysis, and transfusion-related acute lung injury (TRALI) [63]. More specifically, the adverse reactions may be classified based on the type of the reaction, i.e., immune, physicochemical and infectious. Immune-mediated reactions involve allergic and anaphylactic reactions, hemolytic events and TRALI. Physicochemical events include transfusion-associated circulatory overload (TACO) and reactions to additives. Infectious reactions may be due to contamination by bacteria, viruses, protozoa or due to the variant Creutzfeldt-Jakob disease (vCJD) [63,64]. However, given that plasma adverse effects are very unlikely following the recommended procedures for obtaining and administering it, and given the benefits of its administration, and more specifically, of its early administration in B-cell-depleted patients, as can be seen in the many cases of persisting COVID-19 that were cured with CP after its administration, there is good evidence on the use of CP to treat immunocompromised patients with COVID-19 [62,65,66].

This study has some notable limitations. First of all, the sample is small, meaning that the results should be read with caution, while future studies with larger samples of patients should be performed to allow safe conclusions to be drawn. Furthermore, the results of the multivariate analysis may be affected by the small sample, thus, if the sample was larger, the effect of the CP on re-admission could have been statistically significant. Moreover, some data were missing, for example the presence of respiratory failure on presentation for two out of the thirty-nine patients. Finally, there is a risk of bias, as in many studies with CP, as the patient population involves patients with multiple morbidity and a high risk for mortality, thus, the results should be read cautiously.

## 5. Conclusions

Patients with previous treatment with B-cell-depleting monoclonal antibodies hospitalized for COVID-19 have a similar mortality as patients in the general population. Patients who died had a longer duration of hospitalization as confirmed by a multivariate logistic regression analysis. Treatment with CP in this patient population may be associated with lower rates for readmission for COVID-19, even though this was not identified to be independently associated with a lower likelihood for readmission in our multivariate logistic regression analysis model. Further studies should be performed to identify the role of CP in patients with treatment with B-cell-depleting monoclonal antibodies suffering from COVID-19.

## Figures and Tables

**Table 1 viruses-15-00756-t001:** Patients’ characteristics in total and in regard to whether they received treatment with plasma.

Characteristic	All Patients (n = 39)	Received Plasma (n = 21)	Did Not Receive Plasma (n = 18)	*p*
Age, years, mean (SD)	66.3 (12.8)	68.5 (13.5)	63.7 (11.8)	0.2461
Male gender, n (%)	20 (51.3)	10 (47.6)	10 (55.6)	0.7512
Diagnosed until December 2021 *	17 (43.6)	9 (42.9)	8 (44.4)	1
Diagnosed after December 2021 **	22 (56.4)	12 (57.1)	10 (55.6)	1
Respiratory failure on presentation, n (%) ***	10 (27)	5 (26.3)	4 (22.2)	1
Fever, n (%)	26 (66.7)	11 (52.4)	15 (83.3)	0.0508
Infiltrates in chest X-ray, n (%)	24 (61.5)	12 (57.1)	12 (66.7)	0.7424
IgG levels, mg/dL, median (IQR)	551.5 (468.3–662.3)	584 (471–706.5)	534 (421–734.5)	0.7228
Lymphocytes, cells/μL, median (IQR)	600 (400–1000)	700 (450–1050)	600 (400–1025)	0.6394
Hematological disease, n (%)	22 (56.4)	13 (61.9)	9 (50)	0.5279
Rheumatological disease, n (%)	15 (38.5)	7 (33.3)	8 (44.4)	0.5254
Number of plasma units used, median (IQR)	NA	3 (2.5–3.5)	NA	NA
Remdesivir, n (%)	35 (89.7)	18 (85.7)	17 (94.4)	0.6094
Corticosteroids, n (%)	37 (94.9)	20 (95.2)	17 (94.4)	1
Antibiotics, n (%)	35 (89.7)	19 (90.5)	16 (88.9)	1
Co-infection, n (%)	4 (10.3)	2 (9.5)	2 (11.1)	1
ICU stay, n (%)	6 (15.4)	1 (4.8)	5 (27.8)	0.0775
Duration of stay, days, median (IQR)	9.5 (5–16.3)	9 (5-14.5)	12 (5.5–28)	0.3164
In-hospital mortality, n (%)	6 (15.4)	3 (14.3)	3 (16.7)	1
Re-admission for COVID-19, n (%)	9 (25)	2 (11.1)	7 (46.7)	0.0469
Positive PCR for SARS-CoV-2 three months after diagnosis, n (%)	5 (12.8)	3 (14.3)	2 (11.1)	1

ICU: intensive care unit; IQR: interquartile range; NA: not applicable; PCR: polymerase chain reaction; SARS-CoV-2: severe acute respiratory syndrome coronavirus 2; SD: standard deviation. *: most likely to be caused by the Alpha, Beta and Delta variants. **: most likely to be caused by the Omicron variant. ***: data were missing for two patients.

**Table 2 viruses-15-00756-t002:** Patients’ characteristics in regard to mortality.

Characteristic	Survived (n = 33)	Died (n = 6)	*p*
Age, mean (SD)	64.7 (12.5)	74.7 (12)	0.0792
Male gender, n (%)	17 (51.5)	3 (50)	1
Diagnosed until December 2021 *	14 (42.4)	3 (50)	1
Diagnosed after December 2021 **	19 (57.6)	3 (50)	1
Respiratory failure on presentation, n (%) ***	8 (25)	1 (20)	1
Fever, n (%)	22 (66.7)	4 (66.7)	1
Infiltrates in chest X-ray, n (%)	21 (63.6)	3 (50)	0.6580
IgG levels, mg/dL, median (IQR)	559 (488.8–750.8)	499.5 (430–569)	0.5385
Lymphocytes, cells/μL, median (IQR)	600 (450–950)	1200 (275–7675)	0.5995
Hematological disease, n (%)	16 (48.5)	6 (100)	0.0267
Rheumatological disease, n (%)	15 (45.5)	0 (0)	0.0649
Received plasma, n (%)	18 (54.5)	3 (50)	1
Remdesivir, n (%)	30 (90.9)	5 (83.3)	1
Corticosteroids, n (%)	31 (93.9)	6 (100)	1
Antibiotics, n (%)	29 (87.9)	6 (100)	1
Co-infection, n (%)	3 (9.1)	1 (16.7)	0.5025
ICU stay, n (%)	3 (9.1)	3 (50)	0.0359
Duration of stay, days, median (IQR)	9 (5–15.8)	28 (7.3–49.5)	0.0588
Re-admission for COVID-19, n (%)	9 (27.3)	NA	NA

ICU: intensive care unit; IQR: interquartile range; NA: not applicable; SD: standard deviation. *: most likely to be caused by the Alpha, Beta and Delta variants. **: most likely to be caused by the Omicron variant. ***: data were missing for two patients.

**Table 3 viruses-15-00756-t003:** Regression analysis of mortality.

Characteristic	Univariate Analysis *p*	Multivariate Analysis *p*	OR (95% CI)
Lymphocytes (per cell/μL)	0.0387	0.333	1 (1–1.001)
Duration of stay (per day)	0.0021	0.042	1.08 (1.003–1.163)
ICU stay	0.0097	0.499	2.422 (0.186–31.548)

CI: confidence interval; ICU: intensive care unit; OR: odds ratio.

**Table 4 viruses-15-00756-t004:** Regression analysis of re-admission among patients who survived.

Characteristic	Univariate Analysis *p*	Multivariate Analysis *p*	OR (95% CI)
IgG levels, per mg/dL	0.0012	0.638	0.999 (0.993–1.004)
Convalescent plasma use	0.022	0.123	0.112 (0.07–1.817)

CI: confidence interval; OR: odds ratio.

## Data Availability

The data are available upon reasonable request from the corresponding author.

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
