# Peer review of "Convalescent Plasma Treatment of Patients Previously Treated with B-Cell-Depleting Monoclonal Antibodies Suffering COVID-19 Is Associated with Reduced Re-Admission Rates"

_viruses, 2023, doi:10.3390/v15030756_

Round 1

Reviewer 1 Report

I have read with attention the paper entitled “Convalescent plasma treatment of patients previously treated with B-cell depleting monoclonal antibodies suffering COVID-19 is associated with reduced re-admission rates” by Ioannou P et al.

I have some questions about the paper.

1.     Patients receive convalescent plasma, but in Materials and Methods there is no description of what authors define as convalescent plasma. Who elaborate it?, What is the warranted neutralizing capacity of plasma against SARS-CoV-2?, How do they measure it for every unit of plasma? Was plasma obtained at the beginning of the pandemia or it is being elaborated recently for covering circulating variants of concern? How many mililiters of plasma has every donation? Do you inactivate and use plasma one month after donation or plasma is under quarantine for 4-6 months?

2.     In table 1, I will differentiate patients diagnosed before 2021 Dec 31st of those diagnosed later; as first use to be Wuhan, beta or delta variants, and patients diagnosed later usually are infected by Omicron variant of concern. It is of interest to know the percentage of patients infected by old VOCs and of those infected by Omicron.

3.     In table 1, would be of interest to know how many patients of those treated with or without plasma received tocilizumab or baricitinib.

4.     1/6 patients died of respiratory failure. What were the causes of death for the other 5 patients?

5.     In table 2, 9 patients had re-admissions and were alive; any of the 6 patients who died had been a re-admission for COVID-19 before dying?

6.     Were all 33 patients who survived cured of COVID-19 with a nasopharyngeal PCR negative? Or some of them remained with a chronic infection?

7.     When authors make a multivariate analysis of factors associated to re-admission among patients who survived, may be p be no significant because n is 9. With a higher n signification could arrive in future analysis of the series.

8.     In the epigraph “limitations”, authors say “patients´ follow-up post-hospital discharge was not performed, thus, data on admissions or mortality after discharge were not available”. I don´t understand this comment. Though retrospectively analysed, readmissions and deaths of this cohort of patients have been performed and recorded along time till the last censored visit.

9.     Finally, the “third limitation” has nothing to do with this paper. Those four lines authors talk about endoscopies. They have no sense.

Author Response

Reviewer 1

I have read with attention the paper entitled “Convalescent plasma treatment of patients previously treated with B-cell depleting monoclonal antibodies suffering COVID-19 is associated with reduced re-admission rates” by Ioannou P et al.

I have some questions about the paper.

  1. Patients receive convalescent plasma, but in Materials and Methods there is no description of what authors define as convalescent plasma. Who elaborate it?, What is the warranted neutralizing capacity of plasma against SARS-CoV-2?, How do they measure it for every unit of plasma? Was plasma obtained at the beginning of the pandemia or it is being elaborated recently for covering circulating variants of concern? How many mililiters of plasma has every donation? Do you inactivate and use plasma one month after donation or plasma is under quarantine for 4-6 months?

Response: Thanks for the comment. We have contacted the blood transfusion department of our hospital and added some contributing authors from relevant departments, as their contribution was significant during the whole process of convalescent plasma treatment. We have forwarded this comment as well as another similar one from another reviewer and have added all the required information in the revised manuscript in the methods section. More specifically, one can now see that the revised manuscript at that point has a large paragraph stating specific details about how the plasma was taken from donors, the inclusion criteria, how the levels of the plasma were evaluated, how the plasma was stored, how it was inactivated and how it was tested against pathogens. We now feel that this point in the revised manuscript is clear for the reader and allows him to understand how the convalescent plasma was prepared and administered to the patient.

  1. In table 1, I will differentiate patients diagnosed before 2021 Dec 31st of those diagnosed later; as first use to be Wuhan, beta or delta variants, and patients diagnosed later usually are infected by Omicron variant of concern. It is of interest to know the percentage of patients infected by old VOCs and of those infected by Omicron.

Response: Thanks for the comment. We have added that information in Tables 1 and 2 to allow the reader to see how many patients were infected by older variants of concern. The majority were infected after December 2021. There were no statistically significant differences in terms of convalescent plasma use or mortality.

  1. In table 1, would be of interest to know how many patients of those treated with or without plasma received tocilizumab or baricitinib.

Response: None of these patients received tocilizumab or baricitinib. We have added this information in the revised version of the manuscript.

  1. 1/6 patients died of respiratory failure. What were the causes of death for the other 5 patients?

Response: Thanks for the comment. Actually, respiratory failure was the cause of death in three patients, but only one of them presented with respiratory failure in the context of COVID-19. In the other two who died of respiratory failure, it developed later on, in the context of complications of the hospitalization. Septic shock was the cause of death in the other three patients, again, as a result of complications during the hospitalization. We have added this information in the revised version of the manuscript.

  1. In table 2, 9 patients had re-admissions and were alive; any of the 6 patients who died had been a re-admission for COVID-19 before dying?

Response: One patient who died had been re-admitted for COVID-19 before dying. We have added this information in the revised version of the manuscript.

  1. Were all 33 patients who survived cured of COVID-19 with a nasopharyngeal PCR negative? Or some of them remained with a chronic infection?

Response: Five out of the 33 patients who survived had a positive nasopharyngeal PCR for at least three months after initial diagnosis and were considered to have a chronic infection by SARS-CoV-2. We have added this information in the revised version of the manuscript in the text in the results section and in Table 1 as well.

  1. When authors make a multivariate analysis of factors associated to re-admission among patients who survived, may be p be no significant because n is 9. With a higher n signification could arrive in future analysis of the series.

Response: Indeed. That is correct. We have modified the limitations subsection of the discussion section of the revised manuscript and now this can be seen at that point.

  1. In the epigraph “limitations”, authors say “patients´ follow-up post-hospital discharge was not performed, thus, data on admissions or mortality after discharge were not available”. I don´t understand this comment. Though retrospectively analysed, readmissions and deaths of this cohort of patients have been performed and recorded along time till the last censored visit.

Response: Thanks for mentioning this. As the reviewer noted, this statement is wrong. We used another manuscript as a template, and that was applicable to the other manuscript. We deleted that.

  1. Finally, the “third limitation” has nothing to do with this paper. Those four lines authors talk about endoscopies. They have no sense.

Response: Thanks for the comment. That is correct for the same reason as in comment 8. We have deleted that.

Reviewer 2 Report

In the manuscript entitled „Convalescent plasma treatment of patients previously treated with B-cell depleting monoclonal antibodies suffering COVID-19 is associated with reduced re-admission rates”, the authors aim to solve the inconsistencies in the literature concerning the benefit of convalescent plasma treatment in patients. While the U.S. Food and Drug Administration (FDA) has given emergency authorization for convalescent plasma therapy with high antibody levels to treat COVID-19, the benefit remains unclear. I have several concerns that need further attention before the manuscript can be considered for publication.

1)      Could the authors give further information about the blood donors what criteria were set for donors to be considered for the convalescent plasma treatment.

2)      The general problem of studies with convalescent plasma treatment is the high bias in the patient cohort as most of them have a high mortality risk and are multimorbid, which can be seen in table 2.  This might strongly effect the outcome of the study.

3)      In the table the percentage in the parenthesis are not perfectly fitting to the number of patients effected like in table 2 Respiratory failure. Please carefully check the percentage in the tables.

4)      Could the authors expand the discussion section including the risks of the procedure of convalescent plasma treatment for patients and stress out that this might be last therapeutic intervention for patients. 

Author Response

Reviewer 2

In the manuscript entitled „Convalescent plasma treatment of patients previously treated with B-cell depleting monoclonal antibodies suffering COVID-19 is associated with reduced re-admission rates”, the authors aim to solve the inconsistencies in the literature concerning the benefit of convalescent plasma treatment in patients. While the U.S. Food and Drug Administration (FDA) has given emergency authorization for convalescent plasma therapy with high antibody levels to treat COVID-19, the benefit remains unclear. I have several concerns that need further attention before the manuscript can be considered for publication.

1)      Could the authors give further information about the blood donors what criteria were set for donors to be considered for the convalescent plasma treatment.

Response: Thanks for the comment. We have contacted the blood transfusion department of our hospital and added some contributing authors from relevant departments, as their contribution was significant during the whole process of convalescent plasma treatment. We have forwarded this comment as well as another similar one from another reviewer and have added all the required information in the revised manuscript in the methods section. More specifically, one can now see that the revised manuscript at that point has a large paragraph stating specific details about how the plasma was taken from donors, the inclusion criteria, how the levels of the plasma were evaluated, how the plasma was stored, how it was inactivated and how it was tested against pathogens. We now feel that this point in the revised manuscript is clear for the reader and allows him to understand how the convalescent plasma was prepared and administered to the patient.

2)      The general problem of studies with convalescent plasma treatment is the high bias in the patient cohort as most of them have a high mortality risk and are multimorbid, which can be seen in table 2.  This might strongly effect the outcome of the study.

Response: Indeed, that is correct. We noted that in the limitations subsection of the discussion section of the study.

3)      In the table the percentage in the parenthesis are not perfectly fitting to the number of patients effected like in table 2 Respiratory failure. Please carefully check the percentage in the tables.

Response: Thanks for the comment. It seems like if something is wrong, but it is not. It is because in some cases, some data were missing. This leads to percentages that do not make sense with the numbers at the brackets on top. We have mentioned that in the limitations subsection of the discussion section of the revised version of the manuscript.

4)      Could the authors expand the discussion section including the risks of the procedure of convalescent plasma treatment for patients and stress out that this might be last therapeutic intervention for patients. 

Response: Thanks for the comment. We agree that there are some adverse events that could occur in patients treated with convalescent plasma. Thus, we have changed the discussion section, right before the limitations subsection, and we added a paragraph that mentions these adverse effects and clearly mentions that this treatment should be provided as a last resort in specific patients, at least until further research supports its wider use.

Round 2

Reviewer 1 Report

After reading the comments added by Ioannou to the questions I did about their paper “Convalescent plasma treatment of patients previously treated with B-cell depleting monoclonal antibodies suffering COVID-19 is associated with reduced re-admission rates” (version 2), I think the paper is richer and more instructive now with the information added.

I will like to clarify a little bit some data:

1.     COVID-19 plasma contained antibodies against SARS-CoV-2 measured by ELISA. ¿Were them against the spike or the nucleoprotein of the virus?

2.     Blood donors were categorized as high titers if their ELISA value was over 3, being positive over 1.1. What was the higher value of ELISA measured in donors patients? What is the value at wich measurement of ELISA is saturated? What was the median of ELISA value for the patients, being all over 3?

3.     Have you any correlation between ELISA value of antibodies against SARS-CoV-2 and their neutralizing ability? Have you worked with such convalescent plasma triying to find a correlation between ELISA value and titers of neutralizing activity on cell cultures infected with SARS-CoV-2? If this study is not done, neutralizing ability of plasma with ELISA over 3 is a supposition but not an evidence. 

4.     Authors say plasma was inactivated. That is optimal because it lends to use the donated plasma soon for patients recently infected. How long was the mean decay between plasma obtention and plasma usage?

5.     Three patients died of septic shock. Could you enumerate the pathogens responsible of septic shock in those patients if that information is at disposal?

6.     Six patients died of COVID-19. Had anyone of them been admitted in an episode before by COVID-19, or all them died in their first admission?

7.     In lines 323-325 it is stated “Thus, due to the theoretical concern of development of adverse effects, CP treatment should be reserved only as a last resort treatment in patients in need, at least until further evidence supports its wider use.” 

I don´t agree with this assertion. Plasma adverse effects are very unlikely following recommended criteria for obtaining and administering it. You can see the paper “Joyner MJ, Bruno KA, Klassen SA, Kunze KL, Johnson PW, Lesser ER, et al. Safetyupdate: COVID-19 convalescent plasma in 20,000 hospitalized patients. MayoClin Proc. 2020;95:1888–97.” In fact, the sooner it is administered, the shorter the evolution of COVID-19 in B-cell depleted patients, as can be seen in the many cases of persisting COVID cured with convalescent plasma after its administration “Delgado-Fernández, M.; García-Gemar, G.M.; Fuentes-López, A.; Muñoz-Pérez, M.I.; Oyonarte-Gómez, S.; Ruíz-García, I.; 534 Martín-Carmona, J.; Sanz-Cánovas, J.; Castaño-Carracedo, M.Á.; Reguera-Iglesias, J.M.; et al. Treatment of COVID-19 with 535 Convalescent Plasma in Patients with Humoral Immunodeficiency - Three Consecutive Cases and Review of the Literature. Enferm 536 Infecc Microbiol Clin (Engl Ed) 2022, 40, 507–516, doi:10.1016/j.eimce.2021.01.009.” There is good evidence on the use of convalescent plasma to treat immunocompromised patients with COVID-19. See “Bloch EM et al. Guidance on the use of convalescent plasma to treat immunocompromised patients with COVID-19” in CID, 2023: doi:  10.1093/cid/ciad066.

8.     In the line 347, it is stated “in another multivariate logistic regression analysis model”. I will write “in our multivariate logistic regression analysis model”.

Author Response

After reading the comments added by Ioannou to the questions I did about their paper “Convalescent plasma treatment of patients previously treated with B-cell depleting monoclonal antibodies suffering COVID-19 is associated with reduced re-admission rates” (version 2), I think the paper is richer and more instructive now with the information added.

I will like to clarify a little bit some data:

  1. COVID-19 plasma contained antibodies against SARS-CoV-2 measured by ELISA. ¿Were them against the spike or the nucleoprotein of the virus?

Response: Thanks for the comment. Quantitative ELISA was used to determine the concentration of anti-SARS-CoV-2 IgG against receptor binding domain (RBD) of spike protein S1: EI-2606-9601 Quantivac G ELISA kit (EUROIMMUN, Lübeck, Germany). This can be seen in the methods section of the revised manuscript.

  1. Blood donors were categorized as high titers if their ELISA value was over 3, being positive over 1.1. What was the higher value of ELISA measured in donors patients? What is the value at wich measurement of ELISA is saturated? What was the median of ELISA value for the patients, being all over 3?

Response: All convalescent plasma that were used in the present study, had an ELISA value of 3 or higher. The higher ELISA value of the donors was 10.57 and the median ELISA value for the donors was 6.65. This can be seen in the results section of the revised manuscript.

  1. Have you any correlation between ELISA value of antibodies against SARS-CoV-2 and their neutralizing ability? Have you worked with such convalescent plasma triying to find a correlation between ELISA value and titers of neutralizing activity on cell cultures infected with SARS-CoV-2? If this study is not done, neutralizing ability of plasma with ELISA over 3 is a supposition but not an evidence.

Response: Thanks. In the current study, we were not able to perform any assay on SARS-CoV-2 infected cell culture model correlating the ELISA value and the titers of neutralizing activity due to lack of biosafety level III facility. However, the correlation between a quantitative anti-SARS-CoV-2 IgG ELISA of Euroimmun and neutralization activity has been demonstrated (J Med Virol, 2022;94(1):388-392). In particular, Spearman's correlation analysis demonstrated a strong positive relationship between anti-S1 IgG levels and neutralizing antibody titers (r= 0.819, p < 0.0001). High and low anti-S1 IgG levels were associated with a positive predictive value of 72.0% for high-titer neutralizing antibodies and a negative predictive value of 90.8% for low-titer neutralizing antibodies, respectively. These results substantiate the implementation of the QuantiVac ELISA to assess protective immunity following infection or vaccination. This can be seen in the methods section of the revised version of the manuscript.

  1. Authors say plasma was inactivated. That is optimal because it lends to use the donated plasma soon for patients recently infected. How long was the mean decay between plasma obtention and plasma usage?

Response: The mean decay between plasma collection and plasma usage was 38 days. We have added that information in the results section as can be seen in the revised version of the manuscript.

  1. Three patients died of septic shock. Could you enumerate the pathogens responsible of septic shock in those patients if that information is at disposal?

Response: In patients with septic shock, cultures were negative in two patients, but in one patient, Enteroccoccus faecalis and Pseudomonas aeruginosa were isolated in a culture from pus, and were considered the causative pathogens. This can be seen in the results section of the revised version of the manuscript.

  1. Six patients died of COVID-19. Had anyone of them been admitted in an episode before by COVID-19, or all them died in their first admission?

Response: They all died during their first COVID-19-associated admission. This can be seen in the results section of the revised manuscript.

  1. In lines 323-325 it is stated “Thus, due to the theoretical concern of development of adverse effects, CP treatment should be reserved only as a last resort treatment in patients in need, at least until further evidence supports its wider use.”

I don´t agree with this assertion. Plasma adverse effects are very unlikely following recommended criteria for obtaining and administering it. You can see the paper “Joyner MJ, Bruno KA, Klassen SA, Kunze KL, Johnson PW, Lesser ER, et al. Safetyupdate: COVID-19 convalescent plasma in 20,000 hospitalized patients. MayoClin Proc. 2020;95:1888–97.” In fact, the sooner it is administered, the shorter the evolution of COVID-19 in B-cell depleted patients, as can be seen in the many cases of persisting COVID cured with convalescent plasma after its administration “Delgado-Fernández, M.; García-Gemar, G.M.; Fuentes-López, A.; Muñoz-Pérez, M.I.; Oyonarte-Gómez, S.; Ruíz-García, I.; 534 Martín-Carmona, J.; Sanz-Cánovas, J.; Castaño-Carracedo, M.Á.; Reguera-Iglesias, J.M.; et al. Treatment of COVID-19 with 535 Convalescent Plasma in Patients with Humoral Immunodeficiency - Three Consecutive Cases and Review of the Literature. Enferm 536 Infecc Microbiol Clin (Engl Ed) 2022, 40, 507–516, doi:10.1016/j.eimce.2021.01.009.” There is good evidence on the use of convalescent plasma to treat immunocompromised patients with COVID-19. See “Bloch EM et al. Guidance on the use of convalescent plasma to treat immunocompromised patients with COVID-19” in CID, 2023: doi:  10.1093/cid/ciad066.

Response: Yes. This is correct, and thank you for the comment. We added that during the first round of the revisions upon request of the second reviewer, even though we were very considered about that. We have removed that comment and added another comment encouraging the use of convalescent plasma in immunosuppressed patients with COVID-19, given its safety and its effect in patients with immunosuppression, such as those with previous use of B-cell-depleting monoclonal antibodies as can be seen in the discussion section of the revised manuscript.

  1. In the line 347, it is stated “in another multivariate logistic regression analysis model”. I will write “in our multivariate logistic regression analysis model”.

Response: Thanks for the comment. We have changed that, as requested. It can be seen in the conclusions section of the revised manuscript.